# Unintended Detrimental Effects of the Combination of Several Safety Measures—When More Is Not Always More Effective

**Sebastian Brandhorst *** and **Annette Kluge**

Faculty of Psychology, Ruhr-Universität Bochum, 44801 Bochum, Germany; annette.kluge@rub.de
* Correspondence: sebastian.brandhorst@rub.de; Tel.: +49-234-32-24608

**Abstract:** To ensure safety-related behavior in risky operations, several safety measures, such as safety-related rules and safety management systems including audits, rewards, and communication, are implemented. Looking at each single measure, it is reasonable to assume that each one leads to rule compliance, but how do they interact? In an experimental study, we varied (1) the salience of either safety, productivity, or both, (2) the reward for the compliance and punishment for a violation, (3) the communication of audit results (result- or process-based), and (4) the gain and loss framing of performance indicators. In a 3 × 2 × 2 × 2 factorial between-group design, 497 engineering students in the role of Control Room Operator participated in a five hour simulation of a production year of a chemical plant. Looking at single effects, salient safety goals led to a low number of rule violations compared to the salience of production goals. Interestingly, the interaction of several measures showed that particular combinations of measures were highly detrimental to safety, although altogether, they were assumed to reduce risks. For practice, this means that the effects of safety measures depend on their particular combination and can lead to unwanted effects.

**Keywords:** goal conflict; work-safety tension; safety-related rule violations; behavior-based safety program; safety audits; incentive; punishment; framing; feedback

## 1. Introduction

In Germany alone, four sets of laws deal with occupational safety and health; at the European level, the European Agency for Safety and Health at Work is responsible for drafting directives, guidelines and standards for the member states. At the UN level, the International Labor Organization promotes the safety of workers. However, despite the amount of rules and regulations, events such as the collapse of the Rana Plaza factory in Bangladesh in 2013 with over 1000 deaths, as well as the dam bursts in Brazil in 2015 and 2019 with a total of 278 deaths, suggest that profit goals and safety-related rules and goals seem to be perceived as incompatible. An oil refinery in Cologne (Germany) has a series of accidents with acid rain (four times from 2010 to 2011), leaking aviation fuel that spread underground to 42,000 square meters (2012), 4400 L of hydrocarbon compound leaked into groundwater (2012), four more chemical spills from March to October 2012, an explosion with two injured in 2013 and a Toloul tank explosion in 2014. Between 2018 and 2019, the German National Accident Insurance (DGUV) recorded a largely constant number of 497 fatal accidents at work [1]. The number of reportable accidents is just over 1 million [1]. The events outlined and the accident figures show the tension between productivity and safety. The legal requirement to ensure safety seems to collide with the interest of companies to be as profitable as possible. Is there a solution to this problem?

Many of these incidents and accidents are attributed to safety-related rule violations. Rule violations consist of deliberate non-compliance with known rules and regulations, but there is no malicious intent behind them [2]. There are many reasons for rule violations, based on motivation, leadership or work procedures [3], to name a few. However, rules, regulations and the safety culture itself may also be causes or drivers of safety-related rule

violations [3]. This suggests that more rules and regulations may lead to more problems instead of solving or preventing them. In our study, we looked at contextual conditions such as goal setting or the communication of performance indicators, and at concrete measures such as the consequences (reward and punishment) of rule violations and compliance and the feedback from audits. We wanted to analyze their effect on safety behavior.

The purpose of the study is to show how perceived conflicts between productivity and safety goals affect the behavior of people in working contexts in which safety rules are intended to protect themselves, others, and the environment. Do rewards and punishments for rule compliance or rule violation influence decision-making behavior under goal conflicts? Furthermore, can these decisions be influenced by different forms of behavioral feedback? Within the framework of our study on behavioral variants under goal conflicts, we additionally added the effects of performance indicator framing as loss and gain to investigate the influence of different representations of performance indicator displays on safety-related rule violations with and without goal conflicts. In addition to the different goal settings, the performance indicators are termed as context factors, as they represent organizational context information. The results can close the research gap in specialized safety measure research and show the possible interconnection of measures that are usually studied in isolation from each other.

### 1.1. Work-Safety Tension and Goal Conflicts

Work-safety tension is the perceived tension between safe work design and compliance with organizational performance standards. The state of tension results from the fact that the two goals are experienced as incompatible [4]. Work-safety tension is presumed to be a facet of the safety climate (management safety, coworker safety, work–safety tension, and incentives) [4]. Depending on the sectors investigated, 62–95% of the workers state that they would not be able to perform their tasks in the specified time if they complied with all safety regulations [3]. If workers are in a dilemma between personal safety and organizational productivity, it could be assumed that people are more likely to opt for personal safety rather than put themselves at risk in favor of the organization. However, goals have different values, which we will examine below.

The reactions to a goal conflict are domain-dependent and therefore differ if the safety (or other rules related) behavior is towards persons (coworkers) or the organization [5]. For example, work-safety tension has effects on interpersonal behavior but not on deviant behavior in terms of productivity, that is, behavior towards the organization [6]. This can be explained by the different values of goals. If the job depends on achieving the company's production goals, then organizational and personal goals are mixed up. One's own safety is important, but so is the job. If a concrete experience from the environment has shown that people are more likely to have lost their job than, for example, an arm, the threat of losing their job takes on greater significance. Risk perception is therefore very closely linked to personal accident experience [7,8] (by observation or experience).

### 1.2. Goal Conflicts and Rule Violations

The tension between a company's performance standards and its safety regulations, as described above, puts the focus on the handling of safety regulations and, thus, on safety-related rule violations. A rule violation is a deliberate violation of existing regulations by a person, but without the intention of causing harm [9]. As mentioned above, rule violations can simply be caused by the lack of quality of the rules, or the fact that there are simply too many rules, outdated rules, or incomprehensible rules [3]. Rules have the function of organizational control for the coordination of resources and actions and as a form of knowledge storage [10]. However, they are also a manifestation of organizational routines and thus form the basic building block of organizational change [11].

*1.3. Behavior-Based System Approach*

Organizational change in this context means a permanent development and adaptation of the organization to an organizational environment. In order to bring about this organizational change, also with regard to safety culture and safety practices, a reflection of the existing rules by the persons applying them is necessary [11]. This reflection is an essential part of any comprehensive approach to Behavior-Based Safety Management (BBSM) [12]. This approach to maintain or establish occupational safety is intervention-based, with measures that can be roughly divided into antecedent-based and consequence-based measures [13]. Antecedence measures can be understood as prevention, for example, rules and regulations, training, and checklists. Consequence-based measures refer either to downstream regulatory measures, that is, audits of safety behavior, or to measures in response to safety-critical events. The exemplary list of measures shows that the term "behavior-based" can be somewhat misleading, since it does not mean a purely person-based approach, but can and should always contain system-based elements [2]. Regarding safety measures, studies from the domain of transport research show that some measures in combination can have an adverse, if not the opposite, effect [14,15]. Studies on risky behavior have also shown that safety measures do not always have the desired effect, but in many cases they increase the risk propensity [16], since the increased safety measures create an illusion of safety [17].

As a result, workers' reflection on rules and procedures may also lead to a deliberate decision against these rules and procedures. This means in summary that BBSM measures can also lead to counterproductive effects. For these reasons, the present study combines several safety measures, both antecedence-based and consequence-based. We therefore designed the study to close the research gap with regard to interacting safety measures. The presumed effects of the respective single measures are presented in the following section.

*1.4. Hypothesis*

The overall theoretical assumptions in this study are built on the effect of goal conflicts on rule-based behavior as described above. However, outcome-oriented goals in an organization cannot simply be classified as a measure of a security management strategy. Fundamental decisions of the strategic orientation of an organization are under discussion here. For example, a "Best Practice Guidance [ . . . ] for Major Accident Prevention" issued by the U.S. Chemical Safety and Hazard Investigation Board specifically addresses Directors and Executives [18] are in the most favorable position to resolve the work-safety tension. Since the topic of process safety receives the least attention in board meetings [19], the present study aims to show what kind of positive and negative influences strategic management decisions can have on the rule-based behavior of workers.

**Hypotheses 1 (H1).** *There is a main effect of goal conflict. Under conditions with goal conflicts, more rule violations occur than under conditions without goal conflicts.*

In this respect, as a first measure in the context of outcome-related objectives, we vary the consequence of a safety-related rule violation, making it a consequence-based measure. The question of the effects of reward and punishment has engaged different disciplines for quite some time. Several studies show that the mere term "reward" or "punishment" significantly influences decision-making behavior and even the memory of actual performance [20–24] in favor of risk/effort taking under a punishment condition. In our previous studies, participants were punished for rule violations [25–27]. In the last study, there was no difference between reward and punishment—only a production-oriented goal was depicted [28]. In order to examine the effects of consequences, also taking into account other goals and the absence of goal conflict, the second hypothesis is as follows:

**Hypotheses 2 (H2).** *There is a main effect of consequences. Under conditions with punishment, more rule violations occur than under conditions with rewards for rule compliance.*

In combination with the previous consequences, we consider in this study not only which consequence occurs, but also how the consequences are communicated. A meta-analysis showed that the type of feedback, in combination with the goal, influences behavior [29]. If feedback provides detailed information about the performance, it can support procedural knowledge [30], which, in turn, has proven to be an amplifier for rule-based behavior [31,32]. In a further meta-analysis, it was shown that feedback acceptance is more positive with higher procedural knowledge [33]. This means that, if the feedback contains information about the process, it reminds and reinforces the knowledge of procedures and results consequently in fewer rule violations. Based on this, we formulate our third hypothesis:

**Hypotheses 3 (H3).** *There is a main effect of the type of feedback. Under conditions of result feedback, more rule violations occur than under conditions of process feedback.*

As was previously the case with framing the consequence as a reward or punishment, the communication of performance parameters in terms of gain or loss framing can also increase loss prevention or gain promotion [23]. We are guided here by the framing effect, according to which stronger risk taking is expected in loss framing than in gain framing [34]. The influence of this effect on rule-related behavior has already been investigated in previous studies [26–28]. However, the question of the effect under the influence of various goal conflicts or its absence is also open here. We formulate our fourth hypothesis:

**Hypotheses 4 (H4).** *There is a main effect of framing. Under conditions of loss framing, more rule violations occur than under conditions of gain framing.*

With respect to the identified research gap of interacting safety measures, we derive the exploratory research question of interacting safety measures that lead to counterproductive effects. In this sense, we assume that the safety measures that are hypothesized (Hypotheses H1–H4) to have a violation-reducing effect can neutralize each other or even have a violation-enhancing effect. Due to the fact that we cannot derive an effective combination of the investigated measures on the basis of the known literature, we will investigate this question exploratively and will not formulate an explicit hypothesis.

## 2. Sample, Materials and Methods

To test the hypotheses, a 3 (goal conflict) $\times$ 2 (consequences) $\times$ 2 (type of feedback) $\times$ 2 (framing) design was applied. The calculation of the necessary sample was conducted with G*Power 3.1.9.4 (test family: F test; statistical test: ANOVA: fixed effects, omnibus, one-way; type of power analysis: A priori). For the scheduled 24 conditions of the experiment, G*Power gives a sample size of 552 persons, which corresponds to a cell size of 23 persons per condition, considering an expected effect size of f = 0.25. Based on cost-economic considerations and taking into account the study duration of almost 5 h as well as previous experience, we aimed for a cell size of 20 randomly assigned persons per condition, which corresponds to $N = 480$ persons. A total of 497 persons with complete data sets could be included in the analysis with an average age of $M = 23.12$ years (range 18–50, $SD = 3.62$), of which 167 were female (33.5%). The acquisition mainly took place at the Ruhr University Bochum, Germany, among engineering students. The proportion of women studying engineering was 35.8% in the winter semester 18/19 [35].

During the acquisition as well as at the beginning of the study, the cover story communicated that the study was about a didactic optimization of start-up trainings. Before the start, all participants signed an informed consent form. Up to 50€ could be earned for the successful execution of the training, which was especially developed for the fictitious organization "WaterTech", depending on performance. The apparent performance-related remuneration made the goal conflict personal. The deception about the subject of the experiment and the remuneration was necessary so that the rule-related behavior was not influenced by knowledge about its observation. This procedure was described to the faculty's internal ethics committee and declared as being ethically unobjectionable (No. 189, Approval issued on: 18 February 2015).

### 2.1. Applied Simulation and Experimental Environment

The Waste Water Treatment Simulation WaTrSim [36] was used to represent rule-related behavior in an organizational context with high hazards and, therefore, safety-related rules. The participants served the role of a control room operator. Their task is to start up one of 20 plants of the fictitious company WaterTech by means of a predefined start-up procedure to separate the delivered solvent–water mixtures into the solvent and water components. The task of starting up a plant was chosen to depict a process that is associated with a high risk in practice. This safety-critical phase of plant control is particularly well suited for depicting goal conflicts since plant shutdowns are very costly for companies. Neglecting safety in this phase is particularly effective in increasing productivity. The simulation software covered the domain delivery, homogenization, separation and product storage. In total, the software depicted a simulated production year, divided into a training phase with 10 weeks and a production phase with 48 weeks. The simulated production year was divided into 4 quarters of 12 simulated weeks each. In each week, the system was started up anew. A simulated week lasted 3 min in the training phase and 2 min in the production phase. The simulation software offered two different start-up procedures, of which only one was known to the operator at the beginning of the experiment. The productive procedure consisted of 10 steps (Figure 1) and allowed for a maximum production result, but at the expense of plant safety, as would be learned later in the simulated year. During the simulation of the production year, the operator learned that the simultaneous filling of the mixing Tank R1 generated an explosive gas mixture that favored deflagration. The safe procedure, which would become mandatory after a simulated accident, decoupled the filling of the tank and thus prevented the formation of the gas mixture, but with a total of 13 steps, the operator learned that a safe start up procedure required more start-up time.

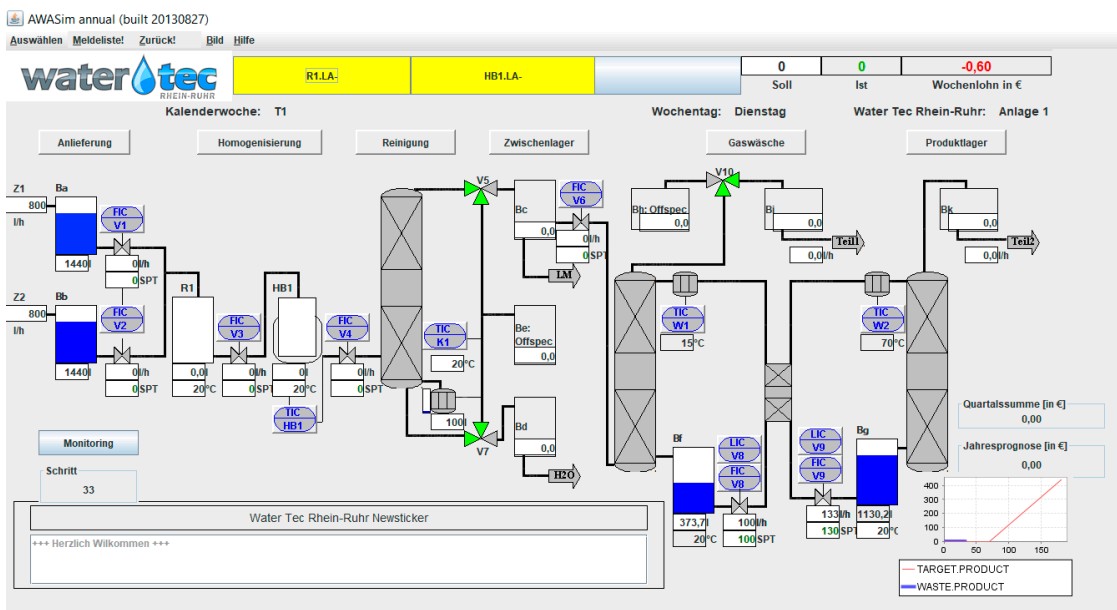

**Figure 1.** AWAsim Interface with valves, tanks and heaters. On the right: (top) numerical salary display, (bottom) graphical production/safety display [37].

In the general experimental set up and the basic version of the experiment, participants as the operators could earn a maximum of 1€ per week if he/she reached the predefined goal (independent of whether it was a safety, production, or combined goal). The payment structure is explained in the following section concerning the independent variables of goal conflicts. Thus, in the best case of rule compliance, in sum, 50€ could be earned at the end of 48 production weeks plus a lump sum of 2€ for the training phase. However,

this amount was not achievable in real terms due to the experimental manipulation of an accident after the first quarter and the subsequently introduced changes in the procedure (from the short to the safe one) and due to possible penalties as a consequence of rule compliance or violation. Thus, the goal conflict became perceptible to the participants and evolved as personally relevant. Table 1 in the section "Goal Conflicts" describes the possible rewards depending on the respective conditions.

The simulated organization's learning after an accident resulted in a change in the mandatory start-up procedure. After the first simulated quarter, an intervention took place in all experimental conditions, in which, for various reasons (see detailed description in the section on independent variables), a modified procedure (see above) was introduced, trained and prescribed by means of operating instructions. After the introduction of the updated rule (depending on which goal setting was realized in the respective condition), compliance with this procedure was monitored by means of software-generated audits. The participants were informed about the audits but not about the frequency of the audits. A total of 9 audits were conducted, 3 in each quarter, but at irregular periods. For the audit, the simulation software checked whether the safety-related mandatory procedure was executed or the risky procedure was used.

### 2.2. Independent Variables

The manipulated factors are the goal conflict (production goal, safety goal and a combination of production and safety goals), the feedback consequence (reward for rule compliance or punishment for rule violation), the type of audit feedback (result and process feedback) and the framing (gain and loss).

#### 2.2.1. Goal Conflicts

The implementation of the goal distinguishes between production, safety and combined goals. The combination means that 50% of the profit comes from production performance and 50% from safety behavior. The background is that, under this condition, no goal conflict is operationalized since compliance with the regulations does not entail any financial disadvantages.

Production goal: Under this condition, the goal conflict is designed with a focus on productivity. This means that the participants were paid according to the amount of treated wastewater; the more they produced, the more they earned. However, the production targets could not be fully achieved in compliance with the safety regulations. To achieve the production goals, participants had to violate the regulations. In the first training, the participants were taught a very productive procedure, with which a maximum income of 1€ per stage could be reached. After the first production quarter (with 12 weeks), due to a fictitious incident, the procedure was extended by 3 action steps, making the procedure safe but less productive. The now safe procedure was communicated by means of a company order as the only valid regulation. In the following 3 quarters, the maximum of 1.00 € could no longer be earned with the safe procedure (only 0.80 €). To counteract the violation of this regulation, the audits described above were used. However, the company still communicated the goal of starting up the plant as quickly as possible and achieving a production target that could not be achieved with the safe procedure.

Safety goal: With the design of the condition of the safety goals, the order of the procedures was reversed, so that first the safe 13-step procedure was trained, and after the intervention, the productive 10-step procedure was taught. In particular, the remuneration modality was changed. From the beginning, the participants were paid for the correct execution of the production steps. The treated amount of waste water was not relevant for the remuneration under this condition. As for generating the process feedback, the simulation software recognized the conducted steps and assigned safety points for each step. As described above, the safety-critical part of the start-up procedure was at the beginning, which is why the conducted steps also give more points. Thus, in total, more safety points

were collected with the safe procedure and converted into monetary units by the underlying compensation model.

Combined production and safety goals: The main purpose of this condition was to eliminate the goal conflict. In the previous conditions, it was not possible to achieve the goals in compliance with the regulations. Under this condition, participants were paid half for production and half for safety. As Table 1 in Column Q2-4 shows, there were no financial disadvantages for full compliance.

### 2.2.2. Consequences

Whether the participants were rewarded for complying with the rule (incentives) or were punished for violating the rule was communicated during the auditing notice. The punishment condition corresponds to the standard sequence of audits as implemented in previous investigations [25–28]. In the case of a negative audit result, the punishment was a reset of the earnings of the week in question to 0.00€. The reward condition, on the other hand, compensated the participants for the loss of earnings, which was reduced by compliance with the regulation. If all audits were positive in the reward condition, the participants received the complete difference value, which otherwise could only be achieved by rule violations.

### 2.2.3. Type of Feedback

The feedback of a conducted audit was either as a pure result description or as a process description. Participants under the condition of result feedback received only information on whether the procedure used corresponded to the specification or whether there was a rule violation. The text output depended on whether the test subjects were under a reward or punishment condition. These are described in the section on incentive conditions. The process feedback listed all the steps actually conducted and made an evaluation with regard to the criteria sequence and operation (setting of expected values). The listed operating steps are highlighted by color:

*Green:* The operation step was performed in the correct order with the correct values.

*Yellow:* The operating step was either performed in the wrong order or with the wrong value.

*Red:* Both the order and set value were wrong.

*Blue:* An operation step was not performed.

The configuration of the stored list and the adjustment considered a marginal tolerance range (if a tank filling of 400 L was expected, the range 350–450 was tolerated) and dependencies of the process steps. The text below the color-coded operating steps provided feedback on whether the applied procedure complied with the regulation or whether it was a rule violation. For example, feedback could be completely green if all steps of the wrong procedure were correctly applied.

### 2.2.4. Framing

It has been widely accepted for some time that the framing of performance indicators has a major impact on how risk-taking or risk-averse people behave. Framing refers to different ways of displaying identical information with respect to a particular reference level [34]. In the present experiment, framing was realized in the design of the training materials as well as on the user interface of WaTrSim. The earnings display (Figure 1, above) played a central role in that respect. According to the above described maximum earnings of 1€ per week, the gain framing showed a positive value of 1€. The loss framing showed the difference from the maximum possible earnings. In the case of the weekly earnings, an earning of 1€ was displayed as −0.00. In the case a participant earned 0.80€, this sum would be displayed as −0.20€ in loss framing. The same principle also applies to the extrapolation of annual earnings. For example, if the extrapolation was 39.45€, this amount would be displayed as −10.55€ in loss framing. This manipulation was ubiquitous due to the implementation on the user interface once the production phase started. In training, it appeared for the first time in the targets and the compensation description.

**Table 1.** Overview table for operationalization of the goal conditions and its implications for procedures (to be executed in WaTrSim) used and earnings.

| | T1 Standard Procedure (6 Training Weeks) | Q1 Earnings Calculation 12 Production Weeks | Intervention | T2 Extended/ Procedure (4 Training Weeks) | Q2-4 |
|---|---|---|---|---|---|
| Production Goal | Valve V9: flow rate 500 L/h<br>Valve V1: flow rate 500 L/h<br>Wait until contents of R1 > 400 L<br>Valve V3: flow rate 1000 L/h<br>Wait until contents of HB1 > 100<br>Switch on heating H1<br>Wait until HB1 > 60 °C<br>Start up column K1<br>Valve V4: flow rate 1000 L/h<br>Valve V6: flow rate 400 L/h | 12 × 1.00€ = 12.00€<br>In each week a maximum of 1€, and thus 12€ per quarter can be earned.<br>In addition 2€ education (training) lump sum are paid. | <u>Incident at WaterTec Rhein-Ruhr</u><br>Deflagration destroys parts of a wastewater treatment plant<br>Addition of a few steps to the standard procedure<br>Prevents the risk of deflagration<br>relatively complex, high lead time<br><u>prevents undesired reaction in mixing Tank R1</u><br>→ safe production | Valve V9: flow rate 500 L/h<br>Deactivate follow-up control<br>Valve V1: flow rate 500 L/h<br>Wait until contents of R1 > 200 L<br>Valve V2: flow rate 500 L/h<br>Wait until contents of R1 > 400 L<br>Valve V3: flow rate 1000 L/h<br>Wait until contents of HB1 > 100<br>Switch on heating H1<br>Wait until HB1 > 60 °C<br>Start up column K1<br>Valve V4: flow rate 1000 L/h<br>Valve V6: flow rate 400 L/h | Reward:<br>V: 36 × 1.00€ − (9 × 0.00) = 36.00€<br>C: 36 × 0.80€ + (9 × 0.80) = 36.00€<br><br>Punishment:<br>V: 36 × 1.00€ − (9 × 1.00) = 27.00€<br>C: 36 × 0.80€ + (9 × 0.00) = 28.80€ |
| Safety Goal | Valve V9: flow rate 500 L/h<br>Deactivate follow-up control<br>Valve V1: flow rate 500 L/h<br>Wait until contents of R1 > 200 L<br>Valve V2: flow rate 500 L/h<br>Wait until contents of R1 > 400 L<br>Valve V3: flow rate 1000 L/h<br>Wait until contents of HB1 > 100<br>Switch on heating H1<br>Wait until HB1 > 60 °C<br>Kolonne K1 in Betrieb nehmen<br>Valve V4: flow rate 1000 L/h<br>Valve V6: flow rate 400 L/h | 12 × 1.00€ = 12.00€<br>In each week a maximum of 1€, and thus 12€ per quarter can be earned.<br>In addition 2€ education (training) lump sum are paid. | <u>Waste water treatment becomes more expensive</u><br>Scarcity of resources forces companies to rethink<br>Standard procedure is shortened by a few steps<br>Increases the productivity of the wastewater treatment plant<br>unsafe gas concentration in Tank R1<br><u>accelerates</u> separation of water and solvent<br>→ increased production | Valve V9: flow rate 500 L/h<br>Valve V1: flow rate 500 L/h<br>Wait until contents of R1 > 400 L<br>Valve V3: flow rate 1000 L/h<br>Wait until contents of HB1 > 100<br>Switch on heating H1<br>Wait until HB1 > 60 °C<br>Kolonne K1 in Betrieb nehmen<br>Valve V4: flow rate 1000 L/h<br>Valve V6: flow rate 400 L/h | Reward:<br>V: 36 × 1.00€ − (9 × 0.00) = 36.00€<br>C: 36 × 0.80€ + (9 × 0.80) = 36.00€<br><br>Punishment:<br>V: 36 × 1.00€ − (9 × 1.00) = 27.00€<br>C: 36 × 0.80€ + (9 × 0.00) = 28.80€ |
| Production and Safety Goal | Valve V9: flow rate 500 L/h<br>Valve V1: flow rate 500 L/h<br>Wait until contents of R1 > 400 L<br>Valve V3: flow rate 1000 L/h<br>Wait until contents of HB1 > 100<br>Switch on heating H1<br>Wait until HB1 > 60 °C<br>Start up column K1<br>Valve V4: flow rate 1000 L/h<br>Valve V6: flow rate 400 L/h | 12 × 1.00€ = 12.00€<br>In each week a maximum of 1€, and thus 12€ per quarter can be earned.<br>No training lump sum. The 2€ are used as a reward. | <u>Incident at WaterTec Rhein-Ruhr</u><br>Deflagration destroys parts of a wastewater treatment plant<br>Addition of a few steps to the standard procedure<br>Prevents the risk of deflagration<br>relatively complex, high lead time<br><u>prevents undesired reaction in mixing Tank R1</u><br>→ safe production | Valve V9: flow rate 500 L/h<br>Deactivate follow-up control<br>Valve V1: flow rate 500 L/h<br>Wait until contents of R1 > 200 L<br>Valve V2: flow rate 500 L/h<br>Wait until contents of R1 > 400 L<br>Valve V3: flow rate 1000 L/h<br>Wait until contents of HB1 > 100<br>Switch on heating H1<br>Wait until HB1 > 60 °C<br>Start up column K1<br>Valve V4: flow rate 1000 L/h<br>Valve V6: flow rate 400 L/h | Reward:<br>V: 50% [36 × 1.00€] + 50% [36 × 0.80€] − (9 × 0.00€) = 28.80<br>C: 50% [36 × 1.00€] + 50% [36 × 1.00€] + (9 × 0.22€) = 38.00<br><br>Punishment:<br>V: 50% [36 × 1.00€] + 50% [36 × 0.80€] − (9 × 1.00€) = 19.80<br>C: 50% [36 × 1.00€] + 50% [36 × 1.00€] + (9 × 0.00€) = 36.00 |

Note: T = Training phase; Q = Quartal; V = Violation; C = Compliance; V1-9 = Valve1-9; R = Mixing tank; K = separation column.

## 2.3. Dependent Variable

For the identification of committed rule violations, the simulation software registered whether the decoupling of V1 and V2 took place. By coupling the valves, the system can recognize whether the 10-step or the 13-step procedure was used. In total, a maximum of 36 rule violations and a minimum 0 rule violations could be committed by participants. The applied procedures in the 1st quarter were not considered, since this took place before the intervention and there was therefore no rule that could be violated.

## 2.4. Demographic and Person-Related Variables

At the beginning of the study, personal variables such as age, sex (m/f), the number of semesters, subject and A-level grade were recorded. In addition, we assessed cognitive variables, including general mental abilities and prior knowledge of wastewater treatment processes. The declarative knowledge about the procedures was measured by means of learning goal control and is listed in Table 2. In addition to goal conflict, the work–safety tension scale measures the experienced tension between work requirements and safety. With the everyday dilemma scale, we recorded how rule-related goal conflicts are dealt with in everyday life. Regarding auditing, we recorded different facets of the perception of the feedback given (evaluation, medium, and result) and its acceptance (feedback acceptance). For a meaningful sequence of questionnaires, the demographic aspects, prior knowledge, and general mental abilities were collected at the beginning of the study. The knowledge tests regarding the trained procedures were administered after the last training round, and the questionnaires on work–safety tension, everyday dilemmas, and feedback perception were administered after the simulation phase. This was to avoid risk sensitization and to equalize concentrated questioning.

**Table 2.** Descriptions of questionnaires used to collect information regarding control variables.

| Variable | Description | Measure | Example Items (Translated, Originally in German) |
|---|---|---|---|
| General mental ability (GMA) (0–50 P.) | General mental ability speed test | Wonderlic test [38] 50 items, 12 min limit | "What is the next number in this series?" 1 - 0,5 - 0,25 - 0,125 - ? |
| Prior knowledge (0–7 P.) | Relevant knowledge about wastewater treatment plants | Self-generated [26,27] 7 items, 1 P. each correct answer | "What does homogenization mean?" |
| Knowledge safe procedure (0–7) | Procedure description with fill-in-the-blanks | Knowledge test, 7 blanks to fill in, 1 P. each correct fill | 4. ▭ |
| Knowledge unsafe procedure (0–5) | Procedure description with fill-in-the-blanks | Knowledge test, 5 blanks to fill in, 1 P. each correct fill | 9. ▭ |
| Work–safety tension (1–5 P.) | Sensing tension between productivity and safety | Work-safety tension [39], 4 items | "Sometimes it is necessary to deviate from safety regulations because of productivity" |
| Everyday dilemma (1–4 P.) | Rule violation in everyday situations | [40], 10 items | "I'd rather risk being caught speeding than be late for an important appointment." |
| Feedback perception (1–5 P.) | Facets of feedback (evaluation, result, medium) | Adapted from Huang et al. [41], 19 items | "I have received sufficient feedback on my activities." |
| Feedback acceptance (1–5 P.) | Behavioral effectiveness of the feedback | Self-generated, 3 items | "I have tried to implement the feedback we received." |

## 2.5. Experimental Procedure

In the following, we describe the general course of the study and *italicize* aspects that occur only in certain conditions for manipulation purposes.

Participants were randomly assigned to certain conditions and had no knowledge of other versions of the experimental design. At the beginning of the study, the participants were welcomed and the inclusion criteria of the study program and language skills were checked. The informed consent form was presented and explained by the experimenter and signed by the participants. The latter was important to ensure that the participants could follow and understand the instructions. Afterwards, the cover story of the trial was presented, according to which the didactic optimization of start-up trainings was discussed. Within the scope of the cover story, participants were informed that they could earn up to 50€, depending on their production success. Any questions would be clarified and the

declaration of consent would be filled out. This was followed by demographic data such as age, sex, course of study, length of study and A-levels. Further questionnaires were filled out, and these are explained in more detail in the section on personal variables. In the subsequent introduction and training phase, the simulation environment with the operating components was described in its sub-areas, "delivery, homogenization, separation, and product storage". This was followed by an explanation of the operating procedure and the training phase. There were two training weeks to practice the adjustment of valves and tanks, two training weeks to practice the whole procedure with a supporting manual, and two training weeks without the manual. *Factor goal conflict—safety goals and combined goals: The calculation of the safety points was explained based on the process feedback, as described above in the "independent variable—goal conflict and type of feedback" section.*

The first quarter was carried out using the standard procedure of the respective goal conflict condition, which is described in Table 1. In the first quarter, the participants completed 12 start-up weeks.

Afterwards, the first break followed, during which the participants were asked to leave the room under a pretext. During this time, the experimenter distributed a fictitious newspaper article describing an incident in one of the plants (for the production and combination goal, Table 1) or a price increase in the preparation process (safety goal, Table 1), which made it necessary to adapt to new requirements and thus to introduce a new procedure. Thus, depending upon the condition, an adapted procedure was presented and trained over 2 weeks with the manual as instructional help and 2 weeks without said manual.

Subsequently, all participants were informed about the following audits to monitor compliance with the modified procedure. Consequence factor: For the punishment conditions, the financial sanctions were communicated if a rule violation was detected. For participants in the reward condition, an additional display was visible and explained in the interface. The so-called "audit balance" showed the sum of the acquired rewards in the case of a positive audit. Feedback factor: Participants in the process feedback received a short explanation of the feedback display, as introduced above in the Process Feedback section.

After that, the 2nd quarter started and was only interrupted by breaks between quarters until the end of the production phase after a total of 58 weeks (10 training sessions and $4 \times 12$ production weeks). At the end of the production phase, the participants were asked to complete a series of questionnaires measuring person-related variables such as feedback acceptance and perceived goal conflict. Meanwhile, according to the respective cover story, the performance-related earnings were determined and the payouts were prepared. In fact, during the processing time of the questionnaires, the investigators merely waited at the presentation desk and tinkled with coins for a little while. Finally, the participants were informed about the actual subject of the experiment and the lump sum of 50€ was paid out. Prior to this, the participants filled out and signed a confidentiality agreement. This was to ensure that information about the purpose of the experiment was not disseminated among the students and had the consequence that a second participation was excluded. During this process, feedback was also obtained from participants, and information relevant for evaluation was written down in a laboratory diary by the experimenter. This was followed by the farewell of the participants and the consolidation of the output files of the WaTrSim program. These were then handed over to the project manager and stored in the safe of the chair in accordance with data protection regulations.

## 3. Results

The data of 522 participants were collected. Of these, 25 were excluded from the analysis because they did not meet the inclusion criterion of a sufficient ability to perform the trained procedure before the start of the production phase. This criterion is defined by the declarative knowledge of the learned procedures, which was assessed by means of a questionnaire before the production phase. Thus, $N = 497$ participants were included in the analysis (age $M = 23.12$ years, range 18–50, $SD = 3.62$, 167 female, Table 3).

**Table 3.** Demographic details of each condition. Age = *M* (*SD*).

| | | Reward | | Punishment | |
|---|---|---|---|---|---|
| | | Result-Feedback | Process-Feedback | Result-Feedback | Process-Feedback |
| Gain | Safety Goals | $N = 20$<br>Sex = 11 f.<br>Age = 22.00<br>(1.65) | $N = 20$<br>Sex = 4 f.<br>Age = 23.35<br>(4.02) | $N = 20$<br>Sex = 9 f.<br>Age = 25.25<br>(7.25) | $N = 21$<br>Sex = 8 f.<br>Age = 23.19<br>(2.38) |
| | Production Goals | $N = 23$<br>Sex = 5 f.<br>Age = 24.39<br>(3.63) | $N = 21$<br>Sex = 8 f.<br>Age = 20.10<br>(1.58) | $N = 20$<br>Sex = 8 f.<br>Age = 23.20<br>(2.61) | $N = 22$<br>Sex = 6 f.<br>Age = 23.00<br>(2.05) |
| | Production & Safety Goals | $N = 18$<br>Sex = 2 f.<br>Age = 21.67<br>(2.89) | $N = 19$<br>Sex = 4 f.<br>Age = 23.74<br>(3.25) | $N = 20$<br>Sex = 8 f.<br>Age = 25.15<br>(4.90) | $N = 21$<br>Sex = 7 f.<br>Age = 22.62<br>(3.60) |
| Loss | Safety Goals | $N = 20$<br>Sex = 6 f.<br>Age = 23.80<br>(2.86) | $N = 21$<br>Sex = 9 f.<br>Age = 22.19<br>(3.09) | $N = 20$<br>Sex = 8 f.<br>Age = 24.45<br>(3.33) | $N = 20$<br>Sex = 5 f.<br>Age = 23.05<br>(3.33) |
| | Production Goals | $N = 22$<br>Sex = 6 f.<br>Age = 22.50<br>(3.04) | $N = 24$<br>Sex = 13 f.<br>Age = 21.88<br>(2.80) | $N = 22$<br>Sex = 6 f.<br>Age = 21.59<br>(2.72) | $N = 23$<br>Sex = 8 f.<br>Age = 24.48<br>(3.53) |
| | Production & Safety Goals | $N = 21$<br>Sex = 11 f.<br>Age = 22.19<br>(3.68) | $N = 20$<br>Sex = 6 f.<br>Age = 24.85<br>(3.57) | $N = 20$<br>Sex = 7 f.<br>Age = 23.75<br>(3.19) | $N = 19$<br>Sex = 2 f.<br>Age = 22.79<br>(4.58) |

*3.1. Hypothesis Testing*

To determine whether the measured control variables can be included as relevant covariates in the analysis, the preconditions were first checked. Due to the correlation with the DV, the variables sex, semester, knowledge of safe and unsafe procedures, feedback acceptance, work–safety tension and everyday dilemmas were considered. Of these, none of them fulfilled the requirement of normally distributed residuals according to the Kruskal–Wallis test. However, due to the central limit theorem and the size of the sample, we assumed that the violation of the condition did not have a significant impact on the results.

Main effects: The test for group differences was performed using ANCOVA in SPSS 24 and adjusted for reported covariates. Regarding the independent variables goal setting (H1: more rule violation with goal conflicts), feedback consequence (H2: more rule violations with punishment than with reward), feedback communication (H3: more rule violations with result feedback than with process feedback), and framing (H4: more rule violations in loss than in gain framing), the ANOVA revealed a significant main effect on goal setting ($F_{(2495)} = 17.49$, $p < 0.01$, $eta^2 = 0.07$). Regarding the analysis results for Hypothesis 1 of the goal conflict, the number of rule violations were found to be significantly different between the condition of the safety goals $M = 2.42$ ($SD = 5.68$), the condition of the production goals $M = 7.08$ ($SD = 10.26$), and the condition with combined goals without conflict $M = 2.80$ ($SD = 6.21$) (Figure 2). The significant difference between these groups corresponds to the expected direction as described in Hypothesis 1, so Hypothesis 1 is considered supported, even though the effect size is considerably small. As an additional analysis, we used combined contrasts to summarize the conditions of the safety goals and the combined goals and compared them to the condition of the production goals. Results show that the number of rule violations were significantly lower in the groups with safety goals ($M = 2.41$, $SD = 5.66$) and combined safety and productivity goals ($M = 2.80$,

*SD* = 6.21) than under the condition with production goals (*M* = 7.04, *SD* = 10.24), with a mean difference of −4.44 (*SE* = 0.84), *p* < 0.001, *d* = 0.68.

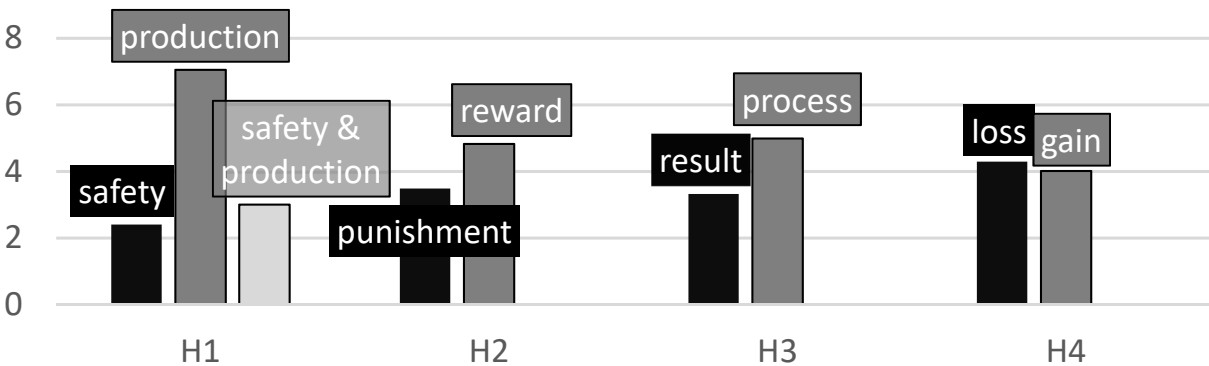

**Figure 2.** Average number of rule violations under the conditions, sorted by Hypotheses H1-H4 for main effects.

No main effect was found for audit consequence ($F_{(1496)}$ = 2.44, *p* = 0.12, $eta^2$ < 0.01), feedback communication ($F_{(1496)}$ = 3.36, *p* = 0.07, $eta^2$ = 0.01), or framing ($F_{(1495)}$ = 0.18, *p* = 0.67, $eta^2$ < 0.01) alone. Hypotheses 2–4 are rejected.

In terms of content, this means that, under all the conditions examined, only the goal setting had a significant effect on the number of safety-related rule violations. Neither the direct measures, such as audit consistency and audit communication, nor framing, in their various forms, led to differences in the number of safety-related rule violations.

Exploratory Analysis of Interaction Effects

With regard to six possible interactions between the four IVs, three relationships were found to have significant interaction effects in the analyses.

(1) The previously reported main effect of goal setting has a significant interaction with framing ($F_{(2495)}$ = 4.70, *p* < 0.01, $eta^2$ = 0.02, Figure 3a). Gain-framing shows more rule violations under the conditions of production (*M* = 7.57, *SD* = 10.71) and combined goals (*M* = 3.93, *SD* = 9.40) compared to loss framing (production: *M* = 6.61, *SD* = 9.84; combination: *M* = 2.49, *SD* = 5.26). In this respect, the rule violations in gain framing decrease in the security target condition (*M* = 0.68, *SD* = 2.36) compared to the rule violations under the loss framing condition (*M* = 4.15, *SD* = 7.27).

(2) The interaction between framing and audit consequence ($F_{(1496)}$ = 5.51, *p* < 0.05, $eta^2$ = 0.01) shown in Figure 3b shows more rule violations in the penalty condition of loss framing (*M* = 4.57, *SD* = 8.41) than in gain framing (*M* = 2.60, *SD* = 6.06), but this is reversed under the reward condition, so more rule violations were found in gain framing (*M* = 5.39, *SD* = 9.88) than in loss framing (*M* = 4.26, *SD* = 7.22).

(3) The interaction between audit consequence and feedback communication ($F_{(1.496)}$ = 6.41, *p* < 0.05, $eta^2$ = 0.01, Figure 3c shows a very similar number of rule violations in the penalty condition for result feedback (*M* = 3.51, *SD* = 7.02) and process feedback (*M* = 3.67, *SD* = 7.74), but in the reward condition there are significantly more rule violations for process feedback (*M* = 6.26, *SD* = 10.13) than for result feedback (*M* = 3.33, *SD* = 6.44).

With regard to our exploratory research question, this means that there are unintended detrimental effects of the combination of safety measures examined in our study.

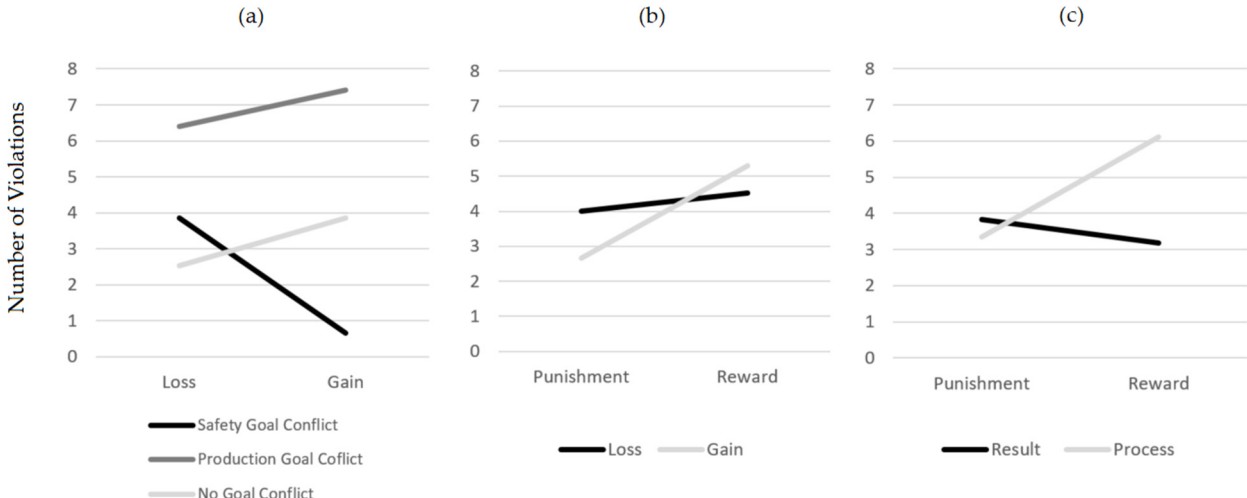

**Figure 3.** Frequencies of rule violations, broken down between the factors of significant interaction effects; (**a**) Interaction between framing and goals, (**b**) interaction between, consequence and framing; (**c**) interaction between consequence and feedback.

### 3.2. Post Hoc

Additionally, we report here the post hoc analysis of the control variables regarding their correlations with the dependent variable rule violations and among each other. Table 4 is organized in clusters of demographic, cognitive and attitude-related variables.

Among the demographic variables, only age and the semester number at the university correlate with each other. With regard to cognitive variables (general mental ability (GMA), prior knowledge, and declarative knowledge), age shows a negative correlation with GMA and declarative knowledge regarding the safe procedure. With regard to attitude-oriented variables (Table 4, Variables 10–13), among the demographic variables, only sex shows a correlation in the sense that men tended to commit more rule violations in everyday life and reported a higher work–safety tension and a lower acceptance of feedback. In contrast, all cognitive-oriented variables show an average correlation among themselves. With regard to attitude-oriented variables, cognitive control variables show a lower work–safety tension with more declarative knowledge about the unsafe procedure. In the case of attitude-oriented variables, the reported work–safety tension and the everyday dilemmas correlate with all other variables except feedback perception.

Compared to demographic or cognitive variables, person-related aspects (e.g., feedback acceptance) are significantly correlated to rule-related behavior. In addition, some cognitive variables are relevant to rule-related behavior. The weakest relation with rule-related behavior can be found in the cluster of demographic variables.

**Table 4.** Correlation matrix between the control variables and the dependent variable of the rule violation, in demography, cognition, and person-related clusters.

| | | 1 | 2 | 3 | 4 | 5 | 6 | 7 | 8 | 9 | 10 | 11 | 12 |
|---|---|---|---|---|---|---|---|---|---|---|---|---|---|
| Demographic | Violation (1) | | | | | | | | | | | | |
| | Sex *** (2) | 0.11 * | | | | | | | | | | | |
| | Age (3) | −0.06 | 0.11 | | | | | | | | | | |
| | A-Level (4) | −0.02 | −0.07 | −0.04 | | | | | | | | | |
| | Term (5) | −0.11 * | −0.02 | 0.45 ** | −0.05 | | | | | | | | |
| Cognitive | General Mental Ability (6) | −0.04 | 0.08 | −0.19 ** | −0.06 | 0.05 | | | | | | | |
| | Prior knowledge (7) | −0.04 | 0.13 ** | −0.05 | −0.04 | 0.15 ** | 0.32 ** | | | | | | |
| | Knowledge safe procedure (8) | −0.14 ** | 0.06 | −0.12 ** | 0.03 | 0.04 | 0.26 ** | 0.16 ** | | | | | |
| | Knowledge unsafe procedure (9) | −0.20 ** | 0.05 | −0.04 | 0.02 | 0.11 * | 0.31 ** | 0.20 ** | 0.38 ** | | | | |
| Person-related | Feedback acceptance (10) | −0.23 ** | −0.16 ** | −0.04 | 0.02 | 0.08 | 0.02 | 0.08 | −0.01 | 0.07 | | | |
| | Work–safety tension (11) | 0.15 ** | 0.14 ** | −0.03 | 0.05 | −0.05 | 0.02 | 0.01 | −0.03 | −0.09 * | −0.22 ** | | |
| | Everyday dilemma (12) | 0.15 ** | 0.13 ** | 0.05 | −0.04 | 0.07 | −0.01 | −0.02 | −0.08 | −0.04 | −0.12 * | 0.33 ** | |
| | Feedback perception (13) | −0.08 | 0.00 | 0.06 | 0.04 | 0.07 | −0.05 | 0.05 | 0.02 | 0.00 | 0.23 ** | 0.02 | 0.00 |

Note: * $p < 0.05$; ** $p < 0.01$; *** Eta, 1 = female, 2 = male.

## 4. Discussion

With the present study, we aimed to investigate the possible detrimental but also positive impact of safety measures and their interactions on safety-related behavior. The results partly reflect our hypotheses, but also contain a few surprises. In the following, we discuss the findings. We also include effect size in the discussion, as they seem to be quite small at the surface. While the experimental design may appear somewhat complex at first glance, we have taken the approach of mapping the complexity of work contexts under laboratory conditions. Since we have already observed interactions, especially counterproductive interactions between safety measures in previous studies [28], a comprehensive view of the effects of different measures is of great importance.

### 4.1. General Discussion

In our approach, we varied the goal setting at the simulated organizational level. The focus of the goals on productivity and on safety played a crucial role, but so did the resolving of this conflict. Our results demonstrate that the context of rule communication is of enormous importance. If safety-related regulations are communicated in the sense of safety, only one-third of rule violations are observed, compared to the situation when communication is oriented towards productivity. There is no difference whether the rules are presented in a safety context (with goal conflict) or whether there is no goal conflict at all. By varying which rule is mandatory and which is prohibited, the results also show that this cannot be the origin of the effect. The common denominator lies in the context of safety-sensitive communication.

On the level of concrete safety measures, neither the audit communication (as process and result feedback) nor the consequences of the audit (reward and punishment) showed a main effect in the observed data on rule-related behavior. Even if the tendency is considered, it is contrary to the expected directions both for feedback and for consequences. These results and tendencies require further investigation, as they are fully counterintuitive.

According to expectations, however, there are interactions between the manipulated factors. These can be structured in such a way that the context factors (goal conflict and framing) interact, and the context factor framing then interacts with the safety measure of audit consequence, and the two measures of audit consequence and audit feedback.

From a preliminary consideration, it can be concluded that goal settings in particular have an impact on rule-based behavior and that safety measures also interact with each other. We will discuss these in the following.

To discuss the interaction of the context factors of goal conflicts and framing, the condition of production goals (as one of the three aspects of the context factor goal conflict) is left out, since they do not interact with the other factor levels of framing (loss and gain). However, the interaction of the safety-sensitive goal conflict and the condition without goal conflict with the framing factors loss and gain seems to create a dilemma.

Loss framing, which in previous studies showed more rule violations than gain framing [26,27], causes more rule violations under conditions with goal conflict than those without goal conflict. Thus, if the performance parameters are framed negatively, (safety-sensitive) goal conflicts will lead to more rule violations than if there were no goal conflicts. However, the effect is reversed if the performance parameters are positively framed (gain framing). Thus, goal conflicts have a reducing effect on rule violations. If there is no goal conflict under gain framing, the rule violations increase.

Regardless of how the results are examined, the context factors of goal conflict and framing remain on the same level from the point of view of safety communication due to the counterproductive interaction. As an intermediate conclusion, however, it can be said that anything is better than a productivity-oriented goal conflict. As a practical implication, this would mean for an organization that there is no mean to meet all needs. Some workers will react with enhanced safety behavior if the performance indicator is positively communicated (gain-framed) but only under a goal conflict. If an organization erases the goal conflict, the framing of the performance indicator becomes important.

The dilemma continues when considering the interaction between the context factor framing and the measure of audit consequence. In loss framing, the observed number of rule violations after reward and punishment is almost constant, but there is a variation in gain framing. In gain framing, significantly fewer rule violations are committed after punishment than after rewards. The level of rule violations after rewards even increases in gain framing above the level in loss framing. Therefore, if one wants to be independent of the effects of the consequence, a negative presentation of performance indicators is required, for example, in term of loss framing. If, however, these are presented in a positive light, our data suggest that punishment for rule violations, not rewards for compliance, should be used.

The problem with rewards becomes even more serious when the interaction with the safety measure of the audit feedback is considered. Under the punishment condition, there is no significant difference between rule violations in result and process feedback. However, if someone rewards for compliance, rule violations are doubled for process feedback and decreased for result feedback.

There is a very complex dynamic in the dependencies of the effects. Gain framing works well, but only under goal conflicts. If an organization implements gain-framed performance indicators, they must consider punishment for violations, not reward for rule compliance. If they punish, they are free from the effects of the audit feedback, but in the case of safety rewards, the communication of the results might have a counterproductive effect.

*4.2. Theoretical Contribution*

As we have already mentioned in our interim conclusions, a very general conclusion can be drawn: different people react differently to different measures. The approach of Behavior-Based Safety Management is therefore called upon by its mandate to increase safety behavior to take interpersonal differences into account. Therefore, we look at common approaches that can explain the conclusion of variable responsivity with respect to contexts and measures.

Even if the Occupational Safety and Health scene is already singing the swansong of risk homeostasis [42], it is still relevant for studies on driving behavior [43] and is increasingly popular in the domain of information security [44,45]. According to the risk homeostasis theory, which is a compensation theory, people have become accustomed to a certain level of risk. If security measures are then taken, the behavior is adjusted (riskier) to maintain the risk level [46]. Accordingly, in view of our data, the question is *which level of risk is acceptable for the individual* and whether he or she adapts their behavior accordingly or whether the measure actually contributes to a reduction in risky behavior.

Common behavioral theories, such as the Theory of Planned Behavior, General Deterrence Theory, and Protection Motivation Theory, focus on behavioral intention as output. The interplay of these theories, which overlap in certain domains, allows us to understand the different responses to the organizational measures reflected in our data. In addition, the approach of the regulatory focus with the characteristics of the promotion and prevention orientation explains the different reactions to the loss and gain framing. Support for the approach of combining several behavioral theories is also reflected in the correlations between the control variables and the dependent variable. Both cognitive aspects (which are mainly represented in the Protection Motivation Theory) and person-related aspects (especially represented in the Theory of Planned Behavior) show clear correlations with the number of rule violations. If a symbiotic theory building for the creation of a Safety Theory of Rule Violation succeeds, combinations of measures could be designed that counteract the dilemmas of the counterproductive effects of safety measures.

Some external influences, such as safety measures in our study, trigger circular or reflective effects on the effects of other measures or facets of personality, which thus yields unpredictable consequences in the complex interaction, as the results of our study show. However, they are unpredictable only because effect interactions are not considered, whether within model sections and with other model sections of theories, moderated

or mediated by individual personality structures. The degree of the individualization of measures is likely to be limited in practical terms, but should be incorporated into modeling at the theoretical level beforehand in order to be perceived by practitioners. The practical implementation then has at least the possibility of orienting itself according to individual personality structures.

### 4.3. Implications for Further Research

There is still the question of why and how these findings will evolve. What are the decisions and strategies behind the actions we can only describe with our present study? Currently we analyze all log data from every participant (every interaction in every second with every component for every person, which are about 100 million data points) to identify the underlying strategies shaping the behavior. By analyzing the present sample the way we did with a previous study [30], we look forward to answering how and why, along with the insights of the present analysis.

A further question that will shape our next investigations and analyses concerns the distribution of committed rule violations during the simulated production year. Since the dependent variable represents a sum value of the entire production year, the question arises as to how individual events, such as audits, influence rule-related behavior in particular. As in previous analyses [37], in the next steps, we will examine aspects such as the occurrence of rule violations with respect to some phenomena, such as the so-called bomb craters effect.

### 4.4. Strength and Limitations of the Study

The presented findings need to be considered in the light of experimental studies under laboratory conditions. The design and sample selection allows a high internal validity and tries to maximize external validity to the working context. At the same time, however, the simulated working environment is thereby cleared of social contexts whose impact on safety-related behavior is undeniable. On the other hand, the personal relevance of the behavior is ensured with the alleged performance-based compensation. However, it was not possible to credibly implement an actual possible source of danger, which is why this idea was dispensed with in the interest of the credibility of the study design. The temporal investment of 5 h (plus travel to and from the study) in the context of the possible chronic, study-related lack of time and the time limitation in the simulation itself generate acute stress in any case, which triggers risk-related behavior [47,48]. Feedback acceptance can also be used as an indication of the effectiveness of the manipulation, that is, the perception of the simulated work environment. The lower the acceptance of feedback, the more rule violations are committed. This direct correlation at the behavioral level indicates that the simulated environment had a direct effect on the behavior of the participants.

Due to the object of investigation, a field study with a manipulation of safety-related behavior is ethically unacceptable and would not be supported by an organization. Thus, the study provides important indications and impulses for the effectiveness of the investigated influencing factors that of course would need to be tested for their potential applicability in an organizational setting.

We see the comparability of the different goal settings with regard to their effects on rule-based behavior as given. This is because the structure of the regulations and the cause of the behavioral obstacles and their consequences are identical. This means, on the one side, that a safety rule is mandatory for production goals, and this reduces earnings; on the other side, a productive rule is mandatory for safety goals, but this also reduces the remuneration for the participants due to the remuneration via the safety behavior: The conflict type is the same, but the content is different. Thus, we can interpret the shown behavior in relation to the conflict contents.

An advantage as well as disadvantage of the study is the above-average complexity. Altogether, scenarios in 24 different variations were compared. The number of independent variables in the analysis also explains the low effect size. With the respective cell size of approximately 20 persons per condition, however, both the effect size and the fact that,

with this low effect size and sample size, the differences can have statistical relevance is remarkable. In addition, the calculation of the contrasts shows a strong effect strength. This supports the assumption that the eta square considerably underestimates the actual effects. Therefore, we are not only satisfied with the methodology, but especially with the content and practical relevance of the study results. In contrast, the study's conscientious and elaborate planning, with over 20 participants per condition and 36 data points per person, constructs a condition from well over 700 decisions. Despite the described limitations, we conclude that we can reasonably be confident that the internal and external validity of the results are sufficient.

The impression of an over-complex study and its potential interference with data quality and participant behavior only arises when, after reading this paper, all the variants and combinations are known. However, the participants did not have this insight. They only knew their respective condition combination, which was logically closed in itself. Therefore, we do not see any disturbing influence of the complexity on the perception of the study and the behavior of its participants.

*4.5. Practical Implications*

First and foremost, those making decisions for or against or even in the design of safety measures should consult the existing evidence. In the sense of evidence-based management, empirical evidence is one of four sources of information. However, the organizations in which the findings from science and research are embedded also face further challenges with the results of this study. The interactivity and complexity of human behavior does not allow for a one-size-fits-all solution. With the same obviousness that products are developed in organizations to make the best use of ground and material capital, a dedicated research facility is needed to make the best use of human capital. The interdisciplinarity of occupational safety must also be reflected in the safety departments and their areas of activity in terms of personnel. The results of the study indicate that a summary of the scientific consensus (in the sense of evidence-based management) is not adequate for a context-sensitive adjustment of safety measures. Questions need to be clarified that take into account the work activity, the social climate in the organization, departments, and teams, how the individual personality structures interact with these climates, and measures that promote occupational safety and their complex interactions. This cosmos is in each case an independent field of research, from which findings defy generalization, but to which general findings can also only be applied with caution. Rather, evidence-based management offers insight into the relevant factors from which an actual appeal can be derived. This then requires a tailored analysis and derivation of measures by business psychologists that are dedicated to occupational safety and health.

**5. Conclusions**

It can be said that unintended detrimental effects of the combination of several safety measures may distort the objective of reducing incidents and accidents at work. The present study has succeeded in highlighting the complexity and multifaceted nature of safety-related measures at different levels. This is the only way to clearly show the multilayered interactions of measures and the complexity of human behavior. In conclusion, it can be stated that there can be no one-size-fits-all measure. On the other hand, the limits of practicability make it worth thinking about whether safety risks can be reduced to the extent that we would like to, considering the variety of reactions to safety measures.

This is another strong argument that safety management requires permanent representation on the board of directors. Moreover, it is clear that gut feelings and subjective assessments cannot do justice to the complexity of the interplay of different measures. The study thus also advocates the implementation of evidence-based management, especially when it comes to human life and the social reputation of the company.

**Author Contributions:** Conceptualization, S.B. and A.K.; Data curation, S.B.; Formal analysis, S.B.; Funding acquisition, A.K.; Investigation, S.B.; Methodology, S.B.; Project administration, S.B. and A.K.; Resources, A.K.; Supervision, A.K.; Writing—original draft, S.B. and A.K.; Writing—review & editing, S.B. All authors have read and agreed to the published version of the manuscript.

**Funding:** This study is funded by the German Research Foundation (KL 2207/2-3).

**Institutional Review Board Statement:** The study was conducted according to the guidelines of the Declaration of Helsinki, and approved by the Institutional Ethics Committee of the Faculty of Psychology, Ruhr-University Bochum (No. 189, Approval issued on: 18 February 2015).

**Informed Consent Statement:** Informed consent was obtained from all subjects involved in the study.

**Data Availability Statement:** The data presented in this study are available on request from the corresponding author.

**Conflicts of Interest:** There is no conflict of interests influencing this research.

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
