# Peer review of "Unintended Detrimental Effects of the Combination of Several Safety Measures—When More Is Not Always More Effective"

_safety, 2020_

Round 1

Reviewer 1 Report

1. The interaction of safety and productivity goals remains an important question. The authors make a good case for systemic conflict between various safety, regulatory, and performance requirements. The experimental design presents an interesting combination of situations within a detailed simulation. The paper is very well written, clear, and has an appropriate level of detail for understanding the rationale and the results. All in all, this is a very good piece of work and I only have a general comment as a suggestion for improvement.

2. The interactions are indeed very interesting and further exploration of their meaning would be helpful. What decision processes might be underlying the findings. For example, what decision or cognitive process might underly gain framing resulting in less violations when there is goal conflict? It is important to unpack these possibilities some more.

3. In addition, the exploration of interactions might also consider the context of the simulation and participants both from a positive and negative point of view. The controlled environment is a positive, but might there be features of the sample or simulation that limit or distort the meaning of the interactions?

4. The outcome measure is a count variable, and the implications of its distribution should be considered in the analyses or the limitations section.

Author Response

  1. The interaction of safety and productivity goals remains an important question. The authors make a good case for systemic conflict between various safety, regulatory, and performance requirements. The experimental design presents an interesting combination of situations within a detailed simulation. The paper is very well written, clear, and has an appropriate level of detail for understanding the rationale and the results. All in all, this is a very good piece of work and I only have a general comment as a suggestion for improvement.

Thank you very much for your appreciative remarks and helpful comments. In the next paragraphs you will find our specific references to your comments and the corresponding implementations.

  1. The interactions are indeed very interesting and further exploration of their meaning would be helpful. What decision processes might be underlying the findings. For example, what decision or cognitive process might underly gain framing resulting in less violations when there is goal conflict? It is important to unpack these possibilities some more.

With this suggestion, you hit exactly the core of further evaluations. Currently, we are analyzing the log files, which have stored every second of every interaction and non-interaction of the test subjects with the software. In this way, we can track down and assign behavioral patterns for each individual person. We have already gained experience with this analysis on the basis of another data set (Brandhorst & Kluge 2016) and are very curious to see to what extent results can be reproduced and what new insights can be gained. To highlight this, we added in the “further research” section:

“There is still a question mark on why and how does these findings evolve. What are the decisions and strategies behind the actions we can only describe with our present study? Currently we analyze all log data from every participant (every interaction in every second with every component for every person, which are about 100 Million data points) to identify the underlying strategies shaping the behavior. With analyzing the present sample the way we did with a previous study [30], we are looking forward to answer the how and why along with the insights of the present analysis.”

  1. In addition, the exploration of interactions might also consider the context of the simulation and participants both from a positive and negative point of view. The controlled environment is a positive, but might there be features of the sample or simulation that limit or distort the meaning of the interactions?

We have gratefully taken your comment as an opportunity to discuss further the occurrence of risk-related behavior (and whether it should be declared as such in a safe environment at all) and have included the following paragraph supported with further evidence in the limitation section:

“On the other hand, the personal relevance of the behavior could be ensured with the alleged performance-based compensation. However, it was not possible to credibly implement an actual possible source of danger, which is why this was dispensed with in the interest of the credibility of the study design. The temporal investment of 5 hours (plus travel to and from the study) in the context of possible study-related chronic lack of time and the time limitation in the simulation itself generate acute stress in any case, which triggers risk-related behavior [47,48].

  1. The outcome measure is a count variable, and the implications of its distribution should be considered in the analyses or the limitations section.

Here, too, your suggestions are in line with our planned research and analysis projects. In order not to further complicate the present work, we have put further analyses, which also concern the distribution of rule violations throughout the year, on the end of the list. To make this intention clear, we added in the "Further research" domain:

“A further question that will shape our next investigations and analysis concerns the distribution of commited rule violations during the simulated production year. Since the dependent variable represents a sum value of the entire production year, the question arises as to how individual events, such as the audits, influenced the rule-related behavior in particular. As in previous analyses [37], in the next steps we will examine aspects such as the occurrence of rule violations with respect to some phenomena, such as the so-called bomb craters effect.”

Reviewer 2 Report

The methodology presented in the article is extremely innovative and relevant to the behavioral study in the area of security.

However, the article is too long, making its reading complex and not very intuitive.

Perhaps due to the complexity of reading associated with the theme, I cannot see if the independent variables could be taken simultaneously, that is, if the individual consciously takes one of the options (goal conflicts, safety goals ...), or if opting for the goal conflit, for example, could no longer opt for another scenario.

The article and the theme are very interesting, I leave to the authors' suggestion the simplification and reduction to the contents considered essential for the transmission of the results of the experiment.

Author Response

The methodology presented in the article is extremely innovative and relevant to the behavioral study in the area of security.

Response: On behalf of the entire research team, I am very grateful for the appreciative words, which also acknowledges the commitment and dedication of all involved.

However, the article is too long, making its reading complex and not very intuitive.

Response: According to the complexity, we have tried hard to keep the balance between detail and comprehensibility, which has unfortunately increased the text length in favor of comprehensibility. We fully agree that the length of the text makes access to the content more difficult. At the same time, however, reviews tend to demand more rather than less information. We try to do justice to this range of tension as best we can.

Perhaps due to the complexity of reading associated with the theme, I cannot see if the independent variables could be taken simultaneously, that is, if the individual consciously takes one of the options (goal conflicts, safety goals ...), or if opting for the goal conflit, for example, could no longer opt for another scenario.

Response: With these comments, you have drawn our attention to a blind spot. The students were randomly assigned to the different conditions and also had no knowledge that there were other variants of the experiment. This was only clarified in the debriefing after the end of the experiment. Of course, it was then no longer possible for them to participate again. We have integrated these aspects into the manuscript as follows:

Sample-Section:

“Based on cost-economic considerations and taking into account the study duration of almost 5 hours as well as previous experience, we have aimed for a cell size of 20 randomly assigned persons per condition, which corresponds to N = 480 persons.

Experimental procedure (beginning):

“Participants were randomly assigned to then conditions and had no knowledge of other versions of the experimental design.”

Experimental procedure (end):

“This is to ensure that information about the purpose of the experiment is not disseminated among the students and has the consequence that a second participation is excluded.”

The article and the theme are very interesting, I leave to the authors' suggestion the simplification and reduction to the contents considered essential for the transmission of the results of the experiment.

Response: In the preparation of the manuscript, we have carefully considered and made cuts and reductions. According to feedback from reviewers so far, however, more information rather than reductions are desired. We try to keep them as compact as possible, so that even with further additions the balance between content, length and complexity can be maintained.

Reviewer 3 Report

In this study, an experiment using a simulation approach was conducted which varied 1) the salience of either safety goals, productivity goals or both, 2) reward for compliance vs. punishment of a violation, 3) the communication of audit results (results vs. process-based) and 4) the gain vs. loss framing of performance indicators. This is a 3 x 2 x 2 x 2 factorial between-group design.  Engineering students (n = 497) participated in a 5-hour simulation of a production year of a chemical plant. Results showed only one main effect: “salient safety goals” led to a low number of rule violations compared to the salience of production goals in these four key factors. The authors suggested that the effects of safety measures depend on their particular combination which can also lead to unwanted effects.

I think it is a unique and interesting study to use simulation to examine/explore these hypotheses. However, I have several concerns and suggestions.

  1. I don’t think the title is accurate. The results cannot provide an answer to the question “Why more is not always more effective?” It showed some particular combination of different safety measures (interventions) can lead to unwanted effects; however, the results did not provide the reasons.
  1. The ideas of this experiment are interesting and relevant; however, the logic for proposing Hypotheses 2-4 is not clear to me. In terms of safety, why should reward be better than punishment? Why should “process-based feedback communication” be better than “results-based feedback”? Why should gain-framing be better than loss-framing of performance indicators?
  1. The simulation program in this experiment is fairly clearly explained and the key constructs are also clearly explained. However, I have a question as to how accurately their simulation actually simulated and measured those experiences for participants. The surprising results suggest that there might be potential errors in the measure and evaluation of key constructs. It is possible that the high complexity of the simulation interfered with the practical relevance of situations presented.
  1. More details are needed of the questionnaire utilized in the study.
  1. Writing style (clarity): The paper itself is well-written at times; parts of the paper are explained clearly, but there are also numerous grammatical mistakes, typos and poor sentence structure which require editing and revision. The clarity of the paper could be improved if edited by a native English speaker.
  1. Theoretical contribution: I like the use of simulation as the approach in this study. However, the only significant finding of this study is “salient safety goals lead to a low number of rule violations compared to the salience of production goals.” All other results are still quite confusing to me. Please provide more details on the theoretical contribution of the study.
  1. Practice implications: more information is needed in terms of practical implementations.
  1. Limitations: more information is needed regarding this study’s limitations.

Author Response

In this study, an experiment using a simulation approach was conducted which varied 1) the salience of either safety goals, productivity goals or both, 2) reward for compliance vs. punishment of a violation, 3) the communication of audit results (results vs. process-based) and 4) the gain vs. loss framing of performance indicators. This is a 3 x 2 x 2 x 2 factorial between-group design.  Engineering students (n = 497) participated in a 5-hour simulation of a production year of a chemical plant. Results showed only one main effect: “salient safety goals” led to a low number of rule violations compared to the salience of production goals in these four key factors. The authors suggested that the effects of safety measures depend on their particular combination which can also lead to unwanted effects.

I think it is a unique and interesting study to use simulation to examine/explore these hypotheses. However, I have several concerns and suggestions.

Response: We very much appreciate your suggestions and constructive advice.  Taking into account your very pointed questions and suggestions, our manuscript has clearly gained in quality.

 I don’t think the title is accurate. The results cannot provide an answer to the question “Why more is not always more effective?” It showed some particular combination of different safety measures (interventions) can lead to unwanted effects; however, the results did not provide the reasons.

Response: Thank you for the precise observation. We have adjusted the title accordingly. It now reads:

Unintended detrimental effects of the combination of several safety measures- When more is not always more effective.”

With respect to this, we also added the section in the “further research” paragraph:

“There is still a question mark on why and how does these findings evolve. What are the decisions and strategies behind the actions we can only describe with our present study? Currently we analyze all log data from every participant (every interaction in every second with every component for every person, which are about 100 Million data points) to identify the underlying strategies shaping the behavior. With analyzing the present sample the way we did with a previous study [30], we are looking forward to answer the how and why along with the insights of the present analysis.”

    The ideas of this experiment are interesting and relevant; however, the logic for proposing Hypotheses 2-4 is not clear to me. In terms of safety, why should reward be better than punishment? Why should “process-based feedback communication” be better than “results-based feedback”? Why should gain-framing be better than loss-framing of performance indicators?

Response: Following your comments, we have re-examined our hypothesis derivation and description. Here we have closed a few logical gaps that may have made the derivation more difficult to follow. In the following, we have listed the additions to the hypothesis description:

Hyp.2: “. Several studies show that the mere term "reward" or "punishment" significantly influences decision-making behavior and even the memory of actual performance [20–24] in favor of risk/effort taking under a punishment condition.”

Hyp.3: “This mean, that if the feedback contains information about the process, it reminds and reinforces the knowledge of procedures and results consequently in fewer rule violations.”

Hyp.4:” We are guided here by the framing-effect, according to which stronger risk taking is expected in loss framing than in gain framing. [34]. The influence of this effect on rule-related behavior has already been investigated in previous studies [26–28].”

The simulation program in this experiment is fairly clearly explained and the key constructs are also clearly explained. However, I have a question as to how accurately their simulation actually simulated and measured those experiences for participants. The surprising results suggest that there might be potential errors in the measure and evaluation of key constructs. It is possible that the high complexity of the simulation interfered with the practical relevance of situations presented.

Response: Considering a similar question from another reviewer, we have already added the following supplement. Furthermore, we have specifically addressed your suggestion of the effectiveness of the simulation:

On the other hand, the personal relevance of the behavior could be ensured with the alleged performance-based compensation. However, it was not possible to credibly implement an actual possible source of danger, which is why this was dispensed with in the interest of the credibility of the study design. The temporal investment of 5 hours (plus travel to and from the study) in the context of possible study-related chronic lack of time and the time limitation in the simulation itself generate acute stress in any case, which triggers risk-related behavior [47,48].”

Feedback acceptance can also be used as an indication of the effectiveness of the manipulation, i.e. the perception of the simulated work environment. The lower the acceptance of feedback, the more rule violations are committed. This direct correlation at the behavioral level indicates that the simulated environment had a direct effect on the behavior of the participants.

We have also considered your comment regarding the possible confounding effects of the comeplexity of the study and included it in our discussion.

“The impression of an over-complex study and its potential interference with data quality and participant behavior only arises when, as after reading this paper, all variants and combinations are known. However, the participants did not have this insight. They only knew their respective condition combination, which was logically closed in itself. Therefore, we do not see any disturbing influence of the complexity on the perception of the study and the behavior of its participants.”

More details are needed of the questionnaire utilized in the study.

Response: With your comment you have helped us to see that the shortening of the manuscript, while reducing the length and complexity, has left out information such as the location and distribution of the questionnaires. We have supplemented this with the following section:

“For a meaningful sequence of the questionnaires, the demographic aspects, prior knowledge and general mental abilities were collected at the beginning of the study. The knowledge tests regarding the trained procedures were administered after the last training round, and the questionnaires on work-safety tension, everyday dilemmas and feedback perception were administered after the simulation phase. This was to avoid risk sensitization and to equalize concentrated questioning.”

Writing style (clarity): The paper itself is well-written at times; parts of the paper are explained clearly, but there are also numerous grammatical mistakes, typos and poor sentence structure which require editing and revision. The clarity of the paper could be improved if edited by a native English speaker.

Response: Thank you for the comment, which sounded very appreciative to the ears of a non-English speaking author. Once the substantive changes in the manuscript are satisfactory to you and the journal, we will give the manuscript to trained editors for linguistic revision.

Theoretical contribution: I like the use of simulation as the approach in this study. However, the only significant finding of this study is “salient safety goals lead to a low number of rule violations compared to the salience of production goals.” All other results are still quite confusing to me. Please provide more details on the theoretical contribution of the study.

Response: We have taken your suggestion as an impulse to diversify the sections in the discussion. Thus, the aspect "Theoretical implication" has been given its own section with the addition below. The domain "Further Research" has been moved entirely to its own section in order to present the two discussion points separately.

“Some external influences, such as safety measures in our study, trigger circular or reflective effects on the effects of other measures or facets of personality, which thus yields unpredictable consequences in the complex interaction. As the results of our study show. However, they are unpredictable only because interactions of effect are not considered, whether within model sections and with other model sections of theories, moderated or mediated by individual personality structures. The degree of individualization of measures is likely to be limited in practical terms, but should be incorporated into modeling at the theoretical level beforehand in order to be perceived by practitioners at all. The practical implementation then has at least the possibility to orient itself according to individual personality structures.”

Practice implications: more information is needed in terms of practical implementations.

Response: In the sense of the already mentioned diversification of the discussion, we have also dedicated a separate section to the "practical implications" and added the following section to this:

“First and foremost, decisions for or against or even in the design of safety measures should consult the existing evidence. In the sense of evidence-based management, empirical evidence is one of four sources of information. However, the organizations in which the findings from science and research are embedded also face further challenges with the results of this study. The interactivity and complexity of human behavior does not allow for a one-size-fits-all solution. With the same obviousness that products are developed in organizations to make the best use of ground and material capital, a dedicated research facility is needed to make the best use of human capital. The interdisciplinarity of occupational safety must also be reflected in the safety departments and their areas of activity in terms of personnel. The results of the study indicate that a summary of the scientific consensus (in the sense of evidence-based management) is not adequate for a context-sensitive adjustment of safety measures.  Questions need to be clarified that take into account the work activity, the social climate in the organization and departments and teams, how the individual personality structures interact with these climates, and additionally with the measures to promote occupational safety, and its complex interactions with each other. This cosmos is in each case an independent field of research, from which findings defy generalization, but to which general findings can also only be applied with caution. Rather, evidence-based management offers an insight into the relevant impact factors than an actual appeal can be derived from it. This then requires a tailored analysis and derivation of measures by business psychologists that are dedicated to occupational safety and health.”

Limitations: more information is needed regarding this study’s limitations.

Response: We have added some aspects to the discussion of limitations, also by suggestion of other reviewers:

“There is still a question mark on why and how does these findings evolve. What are the decisions and strategies behind the actions we can only describe with our present study? Currently we analyze all log data from every participant (every interaction in every second with every component for every person, which are about 100 Million data points) to identify the underlying strategies shaping the behavior. With analyzing the present sample the way we did with a previous study [30], we are looking forward to answer the how and why along with the insights of the present analysis.”

Round 2

Reviewer 1 Report

Thank you for your responsiveness to my suggestions.

Author Response

Thank you for your responsiveness to my suggestions.

Resp.: Thank you very much for your dedication, appreciation and contribution to the improvement of our paper.

Reviewer 2 Report

Changes in the line of the proposed.

Author Response

Changes in the line of the proposed.

Resp.: Thank you very much for taking the effort and discussing our paper so intensively and appreciatively and contributing to its improvement.

Reviewer 3 Report

There are still grammatical mistakes and typos

Author Response

There are still grammatical mistakes and typos.

Resp.: Taking your feedback into account, we have taken advantage of the journal's language check and will submit the corrected version next. Thank you for your feedback and thorough review of our paper.

Round 3

Reviewer 3 Report

The manuscript looks good to me.